# Expression of SARS-CoV-2 Entry Factors in Human Alveolar Type II Cells in Aging and Emphysema

**DOI:** 10.3390/biomedicines9070779

**Published:** 2021-07-06

**Authors:** Chih-Ru Lin, Karim Bahmed, Hannah Simborio, Hassan Hayek, Sudhir Bolla, Nathaniel Marchetti, Gerard J. Criner, Beata Kosmider

**Affiliations:** 1Department of Thoracic Medicine and Surgery, Temple University, Philadelphia, PA 19140, USA; chih-ru.lin@temple.edu (C.-R.L.); karim.bahmed@temple.edu (K.B.); hannah.simborio@temple.edu (H.S.); hassan.hayek@temple.edu (H.H.); Sudhir.Bolla@tuhs.temple.edu (S.B.); Nathaniel.Marchetti@tuhs.temple.edu (N.M.); Gerard.Criner@tuhs.temple.edu (G.J.C.); 2Center for Inflammation, Translational and Clinical Lung Research, Temple University, Philadelphia, PA 19140, USA; 3Department of Physiology, Temple University, Philadelphia, PA 19140, USA

**Keywords:** alveolar type II cells, aging, emphysema, smoking

## Abstract

Alveolar type II (ATII) cells proliferate and restore the injured epithelium. It has been described that SARS-CoV-2 infection causes diffuse alveolar damage in the lungs. However, host factors facilitating virus infection in ATII cells are not well known. We determined the SARS-CoV-2-related genes and protein expression using RT-PCR and Western blotting, respectively, in ATII cells isolated from young and elderly non-smokers, smokers, and ex-smokers. Cells were also obtained from lung transplants of emphysema patients. ACE2 has been identified as the receptor for SARS-CoV-2, and we found significantly increased levels in young and elderly smokers and emphysema patients. The viral entry depends on TMPRSS2 protease activity, and a higher expression was detected in elderly smokers and ex-smokers and emphysema patients. Both ACE2 and TMPRSS2 mRNA levels were higher in this disease in comparison with non-smokers. CD209L serves as a receptor for SARS-CoV-2, and we found increased levels in ATII cells obtained from smokers and in emphysema patients. Also, our data suggest CD209L regulation by miR142. Endoplasmic reticulum stress was detected in ATII cells in this disease. Our results suggest that upregulation of SARS-CoV-2 entry factors in ATII cells in aging, smokers, and emphysema patients may facilitate infection.

## 1. Introduction

The alveolar epithelium is the barrier between inhaled air and the underlying tissue and is composed of alveolar type II (ATII) and alveolar type I cells [1]. ATII cells produce and secrete pulmonary surfactant, which has anti-viral and anti-inflammatory properties, decreases surface tension, allows for adequate ventilation, and prevents pulmonary edema. ATII cells also have a stem cell potential, as they proliferate and repair the epithelium after damage. Re-epithelialization is orchestrated principally by ATII cells, and this indicates their critical role in lung function.

The severe acute respiratory syndrome coronavirus 2 (SARS-CoV-2) is the causal agent of the COVID-19 outbreak leading to acute respiratory distress syndrome [2]. Pathological examination of the biopsy samples obtained from deceased COVID-19 patients revealed diffuse alveolar damage in the lungs [3]. However, the mechanism is not well understood, and there is a gap in our knowledge of host factors, cell-cell interactions, and proteases that play a significant role in SARS-CoV-2 infection. SARS-CoV-2 is closely related to SARS-CoV, which was responsible for a global outbreak in 2003 [4]. ACE2 (Angiotensin-Converting Enzyme 2) has been identified as the receptor for SARS-CoV entry into lung epithelial cells [4]. Single-cell RNA sequencing data analysis obtained from human lung cells revealed that 83% of ACE2 is expressed on the apical surface of ATII cells. Homologies between the receptor-binding domain/motif of SARS-CoV-2 and SARS-CoV led to the hypothesis that ACE2 could also serve as an entry receptor for SARS-CoV-2. The spike (S) protein of coronaviruses facilitates viral entry into target cells, and evidence obtained using cell lines indicates that ACE2 mediates infection. Cellular serine protease TMPRSS2 (Transmembrane Serine Protease 2) is employed for S protein priming [5]. CD209L (also called L-SIGN) is a type II transmembrane glycoprotein in the C-type lectin family expressed in ATII cells and has been previously described as a receptor for SARS-CoV [6]. It has been reported that CD147 (known as Basigin or EMMPRIN) mediates the SARS-CoV-2 virus entry in host cells [7]. Studies of SARS-Co-V and SARS-CoV-2 infection also suggest CD147 interaction with the spike protein [7,8].

The endoplasmic reticulum (ER) is responsible for protein folding and post-translational modifications that allow further transport of proteins to the Golgi apparatus [9]. This is highly important in SARS-CoV replication, which also utilizes vesicle trafficking within the host cells’ ER. Moreover, the coronavirus replication–transcription complexes and structural proteins assemble within the host ER. Misfolded proteins are transported from the ER to the cytosol. The ER chaperone GRP78 is known for its role in misfolded protein degradation and the unfolded protein response [10]. It was found to relocalize to the cell surface and served as an attachment point for Middle East respiratory syndrome coronavirus (MERS-CoV) and bat coronavirus HKU9.

At the time of writing this manuscript, there have been over 177,515,339 global cases and 3,843,997 deaths caused by COVID-19 [11]. Epidemiology data has demonstrated that the rate of hospitalization or death from COVID-19 is low among children, amidst caution that they can still be infected and transmit the virus [12]. Young age was also correlated with asymptomatic or mild manifestations of COVID-19 [13]. Elderly patients with COVID-19 are more likely to progress to severe disease [14,15]. It has been reported that the proportion of deaths in patients over 60 years old accounts for 81% of the total deaths, which implies that aged people are more vulnerable to the SARS-CoV-2 [16]. Moreover, analysis of lung tissue using RNA sequencing revealed the significant effect of smoking on ACE2 pulmonary expression [17]. Another study showed that smoking led to the upregulation of ACE2 in small airway epithelial cells [18]. Also, smokers had higher odds of progression in COVID-19 severity than non-smokers [19]. However, these studies included patients who had already developed this disease, so the risk estimate does not represent the effect of smoking on contracting COVID-19 in the general population. Furthermore, chronic obstructive pulmonary disease (COPD) remained a predictor of death [17]. Emphysema belongs to COPD and is characterized by airspace enlargement [20]. Cigarette smoke is the main risk factor for this disease development. Comorbidities such as COPD could significantly increase the vulnerability of elderly patients to SARS-CoV-2 infection. The impaired oxygenation caused by COVID-19 brings a great challenge to pulmonary function, which could predict poor outcomes. Smoking, older age, and COPD were identified as risk factors for mortality in patients with COVID-19 [21]. However, there is a gap in our knowledge of the molecular mechanism of their higher susceptibility to SARS-CoV-2 infection and disease severity.

Here, we used ATII cells isolated from young and elderly non-smokers and smoker organ donors. Also, cells were obtained from lung transplants of emphysema patients. To our knowledge, this is the first study using isolated human primary ATII cells to determine the molecular features related to SARS-CoV-2 entry in the context of aging, smoking, and emphysema pathophysiology.

## 2. Materials and Methods

### 2.1. Human Primary ATII Cell Isolation

Lungs were obtained from female and male organ donors whose lungs were not suitable for transplantation and were donated for research from the Gift of Life Donor Program. Donors were selected with a clinical history and X-ray that did not indicate infection, reasonable lung function with a PaO_2_/FIO_2_ ratio of >250, and limited time on a ventilator. Non-smokers never smoked, smokers smoked 10–20 cigarettes per day for at least 3 years, and ex-smokers had quit smoking more than 4 years previously. Tissue was obtained from lung transplants of females and males with emphysema. Young organ donors were 2–28 years old, and the elderly and individuals with this disease were 60–88 years old. ATII cells were isolated as we previously described (*n* = 5–23 per group). Cell purity was determined using SP-C antibody (Santa Cruz Biotechnology, Dallas, TX, USA) by immunofluorescence, as we reported [22,23]. Freshly isolated ATII cells were used for all experiments.

### 2.2. Western Blotting

ATII cells were lysed using mammalian cell PE buffer (G-Biosciences, St. Louis, MO, USA) with protease and phosphatase inhibitors (Gold Biotechnology, St Louis, MO, USA). Western blotting was performed as we previously described [24]. Antibodies against ACE2 (BioRad, Hercules, CA, USA), HERPUD1 (Cell Signaling Technology, Danvers, MA, USA), CD209L (R&D Systems, Minneapolis, MN, USA), TMPRSS2, GRP78, and GAPDH (all from Abcam, Cambridge, UK) were used. HRP-conjugated donkey, anti-rabbit, or anti-mouse IgG (Jackson ImmunoResearch, West Grove, PA, USA) were applied. The blots were developed using Luminata Forte Western HRP Substrate (Millipore, Burlington, MA, USA). The density of bands was quantified using ImageJ and NIH Image software.

### 2.3. RT-PCR

Total RNA was isolated from ATII cells using Quick-RNA MiniPrep (Zymo Research, Irvine, CA, USA) and reverse transcribed into cDNA using the SuperScript IV First-Strand Synthesis System (Thermo Fisher Scientific, Waltham, MA, USA). Gene expression was determined by RT-PCR using the SYBR Green Master Mix (Thermo Fisher Scientific) and StepOnePlus Real-Time PCR System (Applied Biosystems, Waltham, MA, USA). Primers were retrieved from PrimerBank [25] and ordered from Invitrogen (Table 1). The following thermal cycle conditions were used: 95 °C for 10 min, 45 cycles of 95 °C for 15 s, 58 °C for 60 s, and 68 °C for 20 s, followed by 95 °C for 15 s and 60 °C for 15 s. Results were normalized to GAPDH and analyzed using the ΔCt method.

### 2.4. miRNA Analysis

RLT buffer (Qiagen, Hilden, Germany) was applied for ATII cells, followed by miRNAs isolation using miRNeasy Tissue/Cells Advanced Kit (Qiagen, Hilden, Germany). MiRNA was converted into cDNA using the SuperScript IV First-Strand Synthesis System. The miR142 expression was determined by RT-PCR using SYBR Green Master Mix and StepOnePlus Real-Time PCR System. Cycling was performed using the following conditions: 95 °C for 3 min, 40 cycles of 95 °C for 5 sec, and 62 °C for 35 sec. MiR142 homology to the 3′ end of CD209L gene was predicted using TargetScan [26]. Relative fold changes of miR142 expression were normalized to the U6 levels using the ΔCt method. The sequences of the primers are provided in Table 1.

### 2.5. miR142 Overexpression

The human alveolar epithelial A549 cell line was used to analyze the effect of miR142 overexpression on CD209L levels. Cells were cultured in DMEM supplemented with 10% FBS (both from Fisher Scientific) and 100 U/mL penicillin-streptomycin (GE Healthcare Life Sciences, Piscataway, NJ, USA). MiR142 was cloned in pcDNA3.1 plasmid (Addgene, Watertown, MA, USA). DH5α *E. coli* (Thermo Fisher, Waltham, MA, USA) was transformed with control pcDNA3.1 and miR142 plasmids, followed by plasmid DNA isolation using an EasyPrep kit (Bioland Scientific, Paramount, CA, USA). A549 cells were transfected with 2.5 µg plasmid DNA using Polyethyleneimine (PEI) reagent (Polysciences Inc., Warrington, PA, USA) for 24 h. MiR142 overexpression was confirmed by RT-PCR (Table 1) as described above.

### 2.6. Statistical Analysis

To evaluate statistical differences among the groups, a *t*-test or one-way ANOVA (GraphPad Prism Software, version 9, San Diego, CA, USA) was used. A value of *p* < 0.05 was considered significant. Data are shown as the mean ± SEM.

## 3. Results

### 3.1. ACE2 and TMPRRS2 Levels in ATII Cells

Recent clinical data indicate higher susceptibility of smokers, elderly, and COPD patients to COVID-19, although the molecular mechanism is not well known [21]. Therefore, we wanted to compare the SARS-CoV-2 entry gene and protein expression, which may facilitate infection, in human primary ATII cells. We isolated ATII cells from young and elderly non-smokers and smokers, and elderly ex-smokers. ATII cell purity was analyzed by immunofluorescence (Figure 1A). We used Western blotting to determine protein levels. ACE2 expression was significantly increased in elderly smokers compared to young and elderly non-smokers (Figure 1B,C). Moreover, its levels were higher in young smokers than in elderly non-smokers. ACE2 mRNA levels were increased in elderly non-smokers and smokers compared to young organ donors regardless of smoking status (Figure 1D). This gene expression was higher in elderly ex-smokers compared to young non-smokers. TMPRSS2 levels were also analyzed in ATII cells isolated from young and elderly organ donors at the protein and gene levels. Its expression was increased in elderly ex-smokers compared to young organ donors regardless of the smoking status and elderly non-smokers as detected by Western blotting (Figure 1B,C). We also detected increased TMPRSS2 expression in elderly smokers compared to young non-smokers. Moreover, higher TMPRSS2 mRNA levels were observed in elderly non-smokers and ex-smokers than in young organ donors regardless of their smoking status (Figure 1D). Furthermore, its expression was increased in elderly smokers compared to young non-smokers. Our results indicate increased ACE2 and TMPRSS2 levels by smoking and aging.

A SARS-CoV-2 infection causes a higher mortality rate in individuals with COPD [21]. Therefore, we wanted to determine molecular features, which may contribute to COVID-19 severity in patients with emphysema. ATII cells were isolated from non-smokers, smokers, and individuals with emphysema. ACE2 protein expression was higher in ATII cells obtained from smokers and patients with this disease than non-smokers (Figure 2A,B). Also, we detected increased TMPRSS2 protein expression in emphysema patients compared to controls. Both ACE2 and TMPRSS2 mRNA levels were higher in this disease in comparison with non-smokers (Figure 2C). Together, our results suggest that high ACE2 and TMPRSS2 expression in individuals with emphysema may increase the SARS-CoV-2 infection rate.

### 3.2. Regulation of CD209L by miR142 in ATII Cells

CD209L is expressed in ATII cells and was identified as a receptor for SARS-CoV [6]. Our analysis using TargetScan predicted miR142 homology to the CD209L gene (Figure 3A). We overexpressed miR142 in A549 cells (Figure 3B) and analyzed CD209L mRNA (Figure 3C) and protein levels (Figure 3D). Our results indicate that miR142 regulates CD209L expression. Therefore, we wanted to determine CD209L and miR142 expression in ATII cells isolated from young and elderly organ donors and emphysema patients. We found increased CD209L levels and decreased miR142 expression in young smokers compared to non-smokers (Figure 4A,B, Panel I). Also, CD209L protein expression was higher in young smokers than non-smokers (Figure 4C,D, Panel I).

Furthermore, we used ATII cells isolated from the elderly non-smokers, smokers, and ex-smokers for this analysis. CD209L mRNA levels were higher in elderly ex-smokers than in other groups and increased in elderly smokers compared to non-smokers (Figure 4A, Panel II). Analysis of miR142 expression indicated decreased levels in elderly smokers and ex-smokers compared to non-smokers (Figure 4B, Panel II). CD209L expression was also analyzed by Western blotting, and increased levels were detected in elderly smokers and ex-smokers compared to elderly non-smokers (Figure 4C, D, Panel II).

Since cigarette smoke contributes to emphysema development, we wanted to determine CD209L and miR142 expression in ATII cells obtained from non-smokers, smokers, and individuals with this disease. CD209L expression was higher in smokers and emphysema patients than in non-smokers (Figure 4A, Panel III). We found decreased miR142 levels in smokers and individuals with this disease compared to non-smokers (Figure 4B, Panel III). CD209L expression was high in smokers and patients with emphysema as detected by Western blotting (Figure 4C,D, Panel III). Our results indicate increased CD209L gene and protein levels in smokers, the elderly, and emphysema patients and suggest regulation of CD209L by miR142.

### 3.3. Increased GRP78 and HERPUD1 Protein Levels in ATII Cells in Smokers and Emphysema Patients

It has been reported that cigarette smoke induces ER stress, which may facilitate viral infections [10,27]. We used ATII cells obtained from young and elderly non-smokers and smokers and elderly ex-smokers to compare the effect of smoking and aging on GRP78 and HERPUD1 levels. Smoking significantly increased GRP78 expression regardless of age compared to non-smoker organ donors as detected by Western blotting (Figure 5A,B). GRP78 mRNA expression was higher in elderly smokers than young non-smokers (Figure 5C).

HERPUD1 protein levels were increased in elderly smokers compared to young organ donors and elderly non-smokers (Figure 5A,B). Its levels were higher in elderly ex-smokers than young smokers. HERPUD1 mRNA expression was increased in the elderly non-smokers compared to young non-smokers or elderly ex-smokers and upregulated in elderly smokers compared to elderly ex-smokers (Figure 5C).

Furthermore, we determined GRP78 and HERPUD1 protein levels in ATII cells isolated from non-smokers, smokers, and emphysema patients. Higher expression of these proteins was detected in smokers and individuals with this disease than in non-smokers (Figure 6A,B). GRP78 mRNA levels were lower in emphysema patients than in non-smokers and smokers, and HERPUD1 expression was decreased in patients with this disease compared to non-smokers (Figure 6C). Our results suggest that smoking and emphysema increase GRP78 and HERPUD1 protein levels in ATII cells.

### 3.4. High CD147 Expression in ATII Cells in Elderly Smokers

We have recently shown higher CD147 expression in ATII cells isolated from smokers and emphysema patients than non-smokers [23]. Here, we used ATII cells obtained from young and elderly non-smokers, smokers, and ex-smokers to determine the effect of smoking and aging on CD147 expression. Significantly higher CD147 expression was detected in elderly smokers than ex-smokers and non-smokers regardless of age by Western blotting (Figure 7A,B). Also, its levels were higher in young smokers than elderly ex-smokers. Moreover, CD147 mRNA levels were increased in elderly smokers compared to young non-smokers (Figure 7C). Thus, our data suggest a synergistic effect of smoking and aging on CD147 expression.

## 4. Discussion

Autopsy studies of deceased patients with COVID-19 describe the presence of necrosis of alveolar lining cells and ATII cell hyperplasia [3]. However, the lung’s molecular changes in various demographic groups associated with susceptibility to SARS-CoV-2 infections are not well documented. A longitudinal observational study was recently designed to clarify tobacco smoking’s role on COVID-19 severity and progression [28]. We analyzed the expression of genes and proteins related to SARS-CoV-2 entry using ATII cells obtained from young and elderly non-smokers and smokers and emphysema patients to improve our knowledge.

After the SARS-CoV outbreak in 2003, ACE2 was identified as the receptor for entry of the virus into lung epithelial cells [4]. Studies suggest that SARS-CoV-2 also binds ACE2, based on the homologies between a receptor-binding domain/motif, leading to its cellular entry. The need for ACE2 for the virus entry steps mediated by the S protein was confirmed using pseudotyped viruses and SARS-CoV-2 isolated from patients [5,29]. Interestingly, uninfected lungs showed scarce expression of ACE2 in alveolar epithelial cells. Our results indicate higher ACE2 expression in ATII cells obtained from the elderly and smokers, suggesting higher susceptibility of these groups to SARS-CoV-2 infection. A significantly greater number of ACE2-positive alveolar cells in the lungs of patients with COVID-19 than in uninfected controls was observed [3]. Also, we found increased ACE2 protein and gene expression in ATII cells isolated from emphysema patients compared to non-smokers. It was recently shown that ACE2 expression was higher in ATII cells in COPD patients than non-smokers as detected by immunohistochemistry [30]. A previous study reported that human primary ATII cells can be infected with 2003 SARS-CoV [31]. ATII cell infectivity with recombinant SARS-CoV-2-GFP was also shown [32], and analysis of lungs from COVID-19 deceased subjects indicated ATII cell infection [33]. ACE2 expression has been analyzed in lung tissue in healthy non-smokers and smokers and COPD patients based on the transcriptomic data [17]. Results indicate ACE2 upregulation in ever-smokers compared to non-smokers. Also, there was a trend for higher ACE2 levels in COPD, but the results were not consistent. This may be explained by the complexity of lung tissue used for the analysis and the presence of multiple cell types. Our data suggest that ACE2 expression may reflect the permissiveness profile of SARS-CoV-2 and increased risk for viral binding and entry in the lungs, especially in the elderly, smokers, and emphysema patients. Although this may indicate that low ACE2 expression contributes to limiting viral entry, it has been shown that ACE2 is also required for protecting the mice from acute lung injury [34]. Thus, ACE2 inhibition may affect lung repair. On the other hand, the ACE2 downregulation resulting from SARS-CoV binding leads to lung damage and inflammation with leaky pulmonary blood vessels and fibrosis [35]. Importantly, ACE2-derived peptides are potential candidates to inhibit SARS-CoV-2 binding to ACE2 receptors, indicating functional and clinical implications [36].

The S protein of coronaviruses facilitates viral entry into target cells [5]. The cellular serine protease TMPRSS2 is employed for S protein priming, and its activity is essential for viral spread and pathogenesis in the infected host. TMPRSS2 expression was detected in various cell types, including ATII cells, by RNA sequencing [37]. Single-nucleus RNA-sequencing and single-nucleus ATAC-sequencing data indicate that most TMPRSS2 expressing cells were epithelial cells, including ATII cells [12]. We compared TMPRSS2 gene and protein levels in ATII cells isolated from non-smokers, smokers, young, elderly, and emphysema patients. Our results indicate that TMPRSS2 levels were increased in elderly smokers and ex-smokers and individuals with this disease at both protein and mRNA levels. It has been reported that TMPRSS2 mutant mice have reduced the severity of lung pathology after infection with the original SARS-CoV [38].

CD209L was identified as a receptor for SARS-CoV, although it is a much less efficient receptor than ACE2 [6]. Marzi et al. showed that CD209L could augment SARS-CoV infection of already-permissive cells [39]. Our results indicate increased CD209L expression in ATII cells obtained from young smokers compared to non-smokers. Its levels were also higher in elderly smokers, ex-smokers, and emphysema patients than in non-smokers, suggesting the mechanism of their susceptibility to COVID-19. We also detected that miR142, which regulates CD209L expression, was decreased in ATII cells in these groups.

We have previously reported increased CD147 levels in ATII cells isolated from smokers and emphysema patients [23]. Here, we found high CD147 expression in elderly smokers at the protein and gene levels. Notably, an interaction between host cell receptor CD147 and SARS-CoV-2 spike protein has been recently shown, which mediated virus infection [7]. Moreover, the loss of CD147, or blocking CD147, inhibited SARS-CoV-2 replication. On the other hand, CD147 overexpression promoted virus infection. These results indicate that CD147 is a receptor for SARS-CoV-2 infection.

ER stress and sustained unfolded protein response (UPR) signaling are significant contributors to the pathogenesis of several diseases, including inflammatory disorders and viral infections, and can increase their severity [40]. Viruses can interact with the host UPR to maintain an environment favorable for the establishment of persistent infection. We and others reported that cigarette smoke induces ER stress which activates the UPR [27,41]. Prior exposure to cigarette smoke enhanced the UPR in animals infected with the respiratory syncytial virus compared to infected animals exposed to room air [27]. Therefore, smokers and COPD subjects’ lungs are likely to be more sensitive to viral infection-induced ER stress, which may further impact disease progression. GRP78, a stress-inducible chaperone with critical functions in the ER, has been reported to facilitate viral entry for a wide variety of viruses, including coronaviruses [10]. We found increased GRP78 protein levels in young and elderly smokers and high GRP78 mRNA expression in elderly smokers. HERPUD1 is another UPR quality control protein, and its levels rise significantly in response to various stressors [42]. HERPUD1 expression at protein and gene levels was upregulated in elderly smokers. Moreover, both GRP78 and HERPUD1 levels were higher in emphysema patients than non-smokers, as detected by Western blotting. Together, our results suggest ER stress in aging, smokers, and individuals with this disease, which may contribute to their susceptibility to SARS-CoV-2 infection. 

In conclusion, our results show a molecular profile of critical factors involved in SARS-CoV-2 entry using human ATII cells isolated from various groups. The virus has spread to all continents; therefore, a multicenter effort to characterize molecular factors contributing to SARS-CoV-2 infection in populations with diverse genetic backgrounds is essential. Moreover, identifying novel host determinants of pathogenesis will help to develop drug targets. Furthermore, physiologically relevant studies are needed to assess therapeutic agents to treat COVID-19.

## Figures and Tables

**Figure 1 biomedicines-09-00779-f001:**
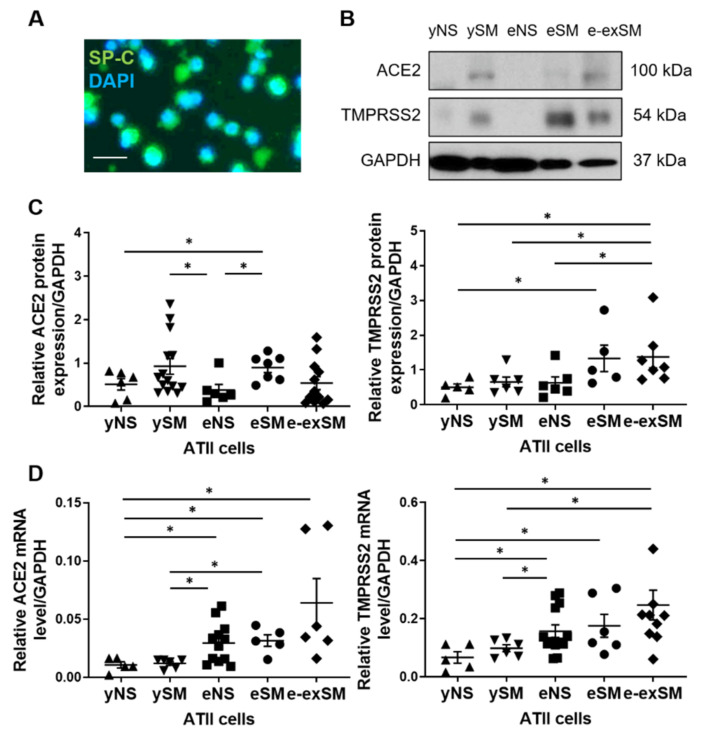
ACE2 and TMRPSS2 expression in ATII cells in young and elderly non-smokers, smokers, and ex-smokers. Freshly isolated ATII cells from lungs obtained from young non-smokers (yNS), young smokers (ySM), elderly non-smokers (eNS), elderly smokers (eSM), and elderly ex-smokers (e-exSM) were used to determine protein and mRNA expression. (**A**) Representative cytospins of freshly isolated ATII cells using SP-C and DAPI by immunofluorescence (*n* = 3), scale bar, 50 µm. (**B**) Western blot images of ACE2 and TMRPSS2 expression (*n* = 5–14 per group). (**C**) Quantification of protein expression normalized to GAPDH. (**D**) ACE2 and TMPRSS2 mRNA levels (*n* = 5–14 per group). * *p* < 0.05. Data are shown as means ± SEM.

**Figure 2 biomedicines-09-00779-f002:**
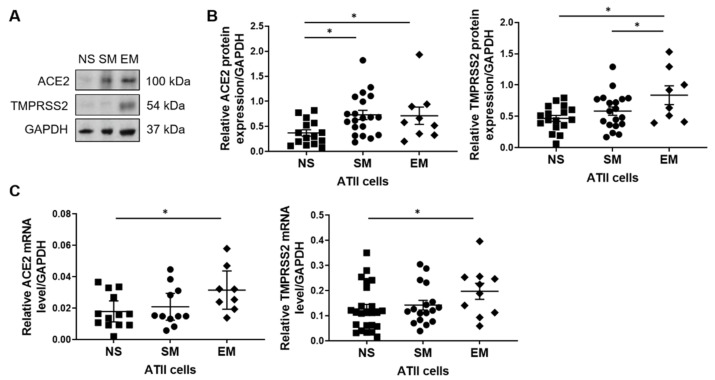
High ACE2 and TMRPSS2 expression in ATII cells in emphysema. ATII cells were isolated from non-smokers (NS), smokers (SM), and emphysema patients (EM). (**A**) Representative images of ACE2 and TMRPSS2 expression using Western blotting (*n* = 8–20 per group). (**B**) Densitometric quantification of protein expression normalized to GAPDH. (**C**) ACE2 and TMPRSS2 mRNA levels (*n* = 8–23). * *p* < 0.05. Data are shown as means ± SEM.

**Figure 3 biomedicines-09-00779-f003:**
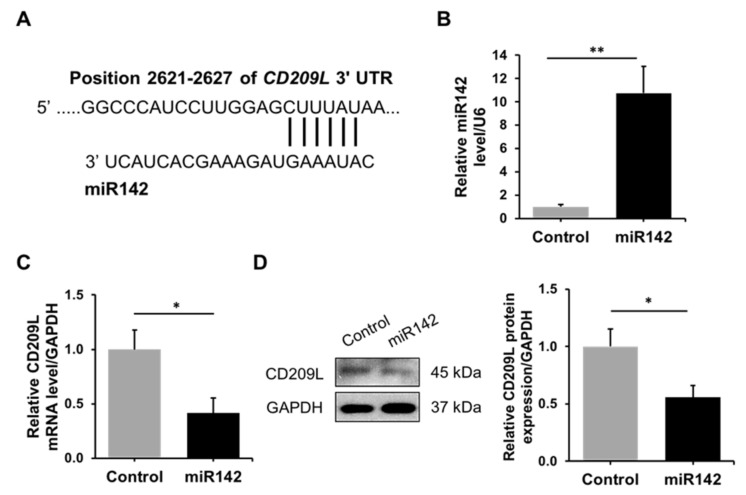
The regulation of CD209L expression by miR142 in A549 cells. (**A**) Predicted homology between CD209L 3′UTR (**top**) and miR142 (**bottom**). (**B**) The miR142 gene was overexpressed in A549 cells, and its levels were confirmed by RT-PCR and normalized to the U6 levels. (**C**) CD209L mRNA expression in A549 cells with miR142 overexpression by RT-PCR. (**D**) CD209L protein levels in A549 cells with miR142 overexpression by Western blotting. Quantification is also shown (*n* = 3). * *p* < 0.05, ** *p* < 0.01. Data are shown as means ± SEM.

**Figure 4 biomedicines-09-00779-f004:**
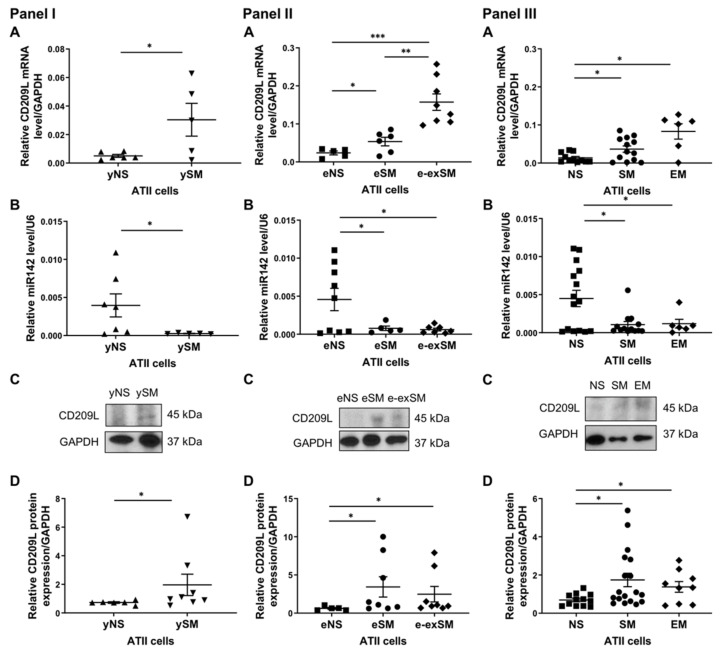
A relationship between miR142 and CD209L in ATII cells. Panel I—ATII cells were isolated from young non-smokers (yNS) and young smokers (ySM). (**A**) CD209L levels determined by RT-PCR (*n* = 5-6 per group). (**B**) miR142 expression (*n* = 5–7 per group). (**C**) CD209L protein expression by Western blotting (*n* = 6–8 per group). (**D**) Quantification of CD209L protein expression normalized to GAPDH. Panel II—ATII cells were obtained from elderly non-smokers (eNS), elderly smokers (eSM), and elderly ex-smokers (e-exSM). (**A**) CD209L mRNA expression (*n* = 5–8 per group). (**B**) miR142 levels were analyzed by RT-PCR (*n* = 5–9 per group). (**C**) CD209L protein levels were determined by Western blotting (*n* = 5–8 per group). (**D**) Densitometric quantification of CD209L protein expression normalized to GAPDH. Panel III—ATII cells were isolated from non-smokers (NS), smokers (SM), and emphysema patients (EM). (**A**) CD209L expression was analyzed by RT-PCR (*n* = 6–14 per group). (**B**) miR142 levels (*n* = 6–15 per group). (**C**) Representative images of CD209L expression using Western blotting. (**D**) Quantification of protein expression normalized to GAPDH. * *p* < 0.05, ** *p* < 0.001, *** *p* < 0.0001. Data are shown as means ± SEM.

**Figure 5 biomedicines-09-00779-f005:**
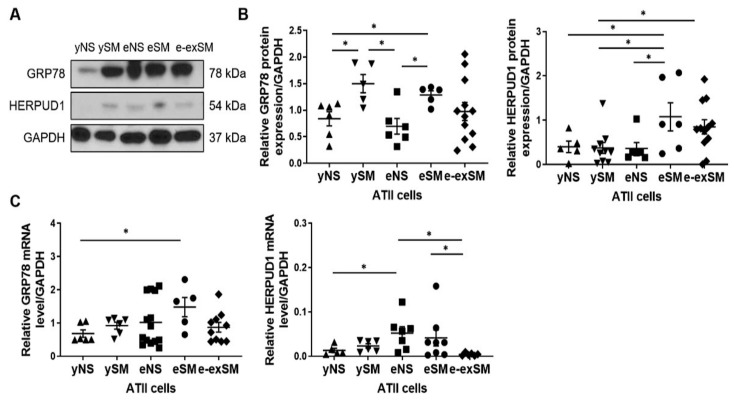
GRP78 and HERPUD1 levels in ATII cells in young and elderly non-smokers, smokers, and ex-smokers. ATII cells were obtained from young non-smokers (yNS), young smokers (ySM), elderly non-smokers (eNS), elderly smokers (eSM), and elderly ex-smokers (e-exSM). (**A**) GRP78 and HERPUD1 expression by Western blotting (*n* = 5–12 per group). (**B**) Densitometric quantification of protein levels normalized to GAPDH. (**C**) GRP78 and HERPUD1 mRNA levels were determined by RT-PCR (*n* = 5–14 per group). **p* < 0.05. Data are shown as means ± SEM.

**Figure 6 biomedicines-09-00779-f006:**
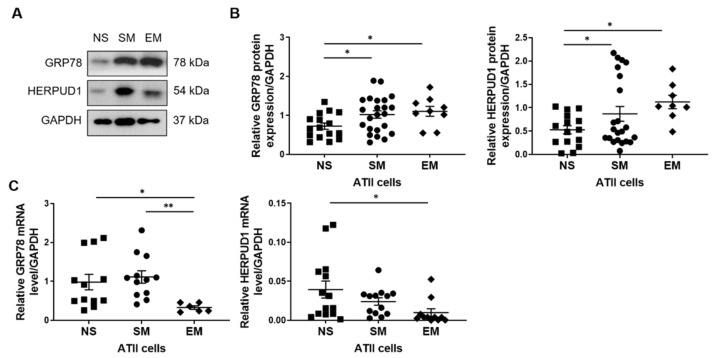
Expression of GRP78 and HERPUD1 in ATII cells isolated from non-smokers, smokers, and emphysema patients. Lungs from non-smokers (NS), smokers (SM), and emphysema patients (EM) were used to isolate ATII cells. (**A**) GRP78 and HERPUD1 protein expression (*n* = 8–23 per group). (**B**) Densitometric quantification of GRP78 and HERPUD1 levels normalized to GAPDH. (**C**) GRP78 and HERPUD1 mRNA levels (*n* = 6-14 per group). * *p* < 0.05, ** *p* < 0.001. Data are shown as means ± SEM.

**Figure 7 biomedicines-09-00779-f007:**
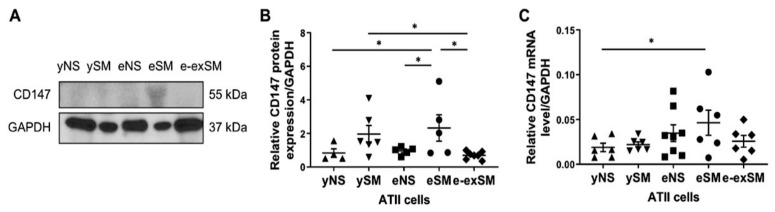
Expression of CD147 in ATII cells isolated from young and elderly non-smokers, smokers, and ex-smokers. ATII cells were isolated from young non-smokers (yNS), young smokers (ySM), elderly non-smokers (eNS), elderly smokers (eSM), and elderly ex-smokers (e-exSM). (**A**) Representative CD147 expression determined by Western blotting (*n* = 4–6 per group). (**B**) Quantification of CD147 expression normalized to GAPDH. (**C**) CD147 mRNA levels by RT-PCR (*n* = 6–8 per group). * *p* < 0.05. Data are shown as means ± SEM.

**Table 1 biomedicines-09-00779-t001:** Sequences of primers used for RT-PCR.

ACE2	F	CGAAGCCGAAGACCTGTTCTA
	R	GGGCAAGTGTGGACTGTTCC
CD147	F	GAAGTCGTCAGAACACATCAACG
	R	TTCCGGCGCTTCTCGTAGA
CD209L	F	TCAAGCAGTATTGGAACAGAGGA
	R	CAGGAGGCTGCGGACTTTTT
GRP78	F	GAAAGAAGGTTACCCATGCAGT
	R	CAGGCCATAAGCAATAGCAGC
HERPUD1	F	ATGGAGTCCGAGACCGAAC
	R	TTGGTGATCCAACAACAGCTT
TMPRSS2	F	CAAGTGCTCCAACTCTGGGAT
	R	AACACACCGATTCTCGTCCTC
GAPDH	F	GGAGCGAGATCCCTCCAAAAT
	R	GGCTGTTGTCATACTTCTCATGG
miR142	F	CGCGCATAAAGTAGAAAGCACT
Common Reverse	R	CGAGGAAGAAGACGGAAGAAT
U6	F	GCTTCGGCAGCACATATACTAAAAT
	R	CGCTTCACGAATTTGCGTGTCAT

Abbreviations: F—Forward, R—Reverse.

## Data Availability

The data supporting the findings of this study are available from the corresponding author upon request.

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
