# Peer review of "Expression of SARS-CoV-2 Entry Factors in Human Alveolar Type II Cells in Aging and Emphysema"

_biomedicines, 2021, doi:10.3390/biomedicines9070779_

Round 1

Reviewer 1 Report

The work analyzed the expression levels of some key entry factors of SARS-CoV-2 human ATII cells from different backgrounds, which is valuable to explain different vulnerability and severity of specific population to SARS-CoV-2 infection. Generally, I'm positive about this work, except some questions/concerns that need the authors to address:

  1. the figures are too small and resolution is too low throughout the manuscript, which needs to be improved very much.
  2. Is it possible to test the infection of SARS-CoV-2 viruses (or  pseudotyped viruses) to the ATII cells from different backgrounds of population that reported in this work, which will be important and greatly improve the significance of this work.
  3. The authors reported the negative correlation between CD209L and miR142, but the direct regulation of CD209L by miR142 need to be validated by cell line study.
  4. Minor: line36-38 lacks reference.   line254: to--->into.

Reviewer 2 Report

An interesting original article about the expression of Sars Cov genes and proteins  in  young and elderly non-smokers, smokers,  ex-smokers and emphisema patients, showing that age, smoking and emphisema may facilitate Covid infection; I have only some minor queries:

In the statistical analysis section, please add what kind of statistical program was used to calculate significance, its maker and location.

A conclusion paragraph should be added, highlighting the future developments of this study.

Thank You

Round 2

Reviewer 1 Report

The manuscript has been revised well. I recommend to accept it as it is.